# Preclinical Therapeutic Efficacy of RAF/MEK/ERK and IGF1R/AKT/mTOR Inhibition in Neuroblastoma

**DOI:** 10.3390/cancers16132320

**Published:** 2024-06-25

**Authors:** Stacey Stauffer, Jacob S. Roth, Edjay R. Hernandez, Joshua T. Kowalczyk, Nancy E. Sealover, Katie E. Hebron, Amy James, Kristine A. Isanogle, Lisa A. Riffle, Lilia Ileva, Xiaoling Luo, Jin-Qiu Chen, Noemi Kedei, Robert L. Kortum, Haiyan Lei, Jack F. Shern, Joseph D. Kalen, Elijah F. Edmondson, Matthew D. Hall, Simone Difilippantonio, Carol J. Thiele, Marielle E. Yohe

**Affiliations:** 1Laboratory of Cell and Developmental Signaling, Center for Cancer Research, National Cancer Institute, NIH, 8560 Progress Drive, Frederick, MD 21701, USA; 2Early Translation Branch, Division of Preclinical Innovation, National Center for Advancing Translational Sciences, 9800 Medical Center Drive, Rockville, MD 20850, USA; jacob.roth@einsteinmed.edu (J.S.R.);; 3Pediatric Oncology Branch, Center for Cancer Research, National Cancer Institute, NIH, 9000 Rockville Pike, Bethesda, MD 20892, USA; 4Department of Pharmacology and Molecular Therapeutics, Uniformed Services University of the Health Services, Bethesda, MD 20814, USArobert.kortum@usuhs.edu (R.L.K.); 5Animal Research Technical Support, Laboratory Animal Sciences Program, Leidos Biomedical Research, Inc., Frederick National Laboratory for Cancer Research, Frederick, MD 21702, USA; 6Small Animal Imaging Program, Laboratory Animal Sciences Program, Leidos Biomedical Research, Inc., Frederick National Laboratory for Cancer Research, Frederick, MD 21702, USA; 7Collaborative Protein Technology Resource, National Cancer Institute, NIH, Bethesda, MD 20892, USA; 8Molecular Histopathology Laboratory, Laboratory Animal Sciences Program, Leidos Biomedical Research, Inc., Frederick National Laboratory for Cancer Research, Frederick, MD 21702, USA

**Keywords:** RAS, MEK, neuroblastoma, IGF1R

## Abstract

**Simple Summary:**

The prognosis for patients with relapsed neuroblastoma is poor, and novel treatment options for these patients are needed. Some relapsed neuroblastoma tumors harbor activating mutations in the RAS/MAPK pathway. In prior studies, single agent MEK or IGF1R inhibitors induced transient responses as single agents in neuroblastoma models. In this study, we tested the efficacy of a combination of the MEK inhibitor trametinib and the IGF1R inhibitor ganitumab in RAS-mutated neuroblastoma models. While the trametinib/ganitumab combination decreased cell viability and tumor growth, the combination did not prevent metastasis of RAS-mutated neuroblastoma. Therefore, further studies on the effect of trametinib and ganitumab on neuroblastoma metastasis are necessary before initiating clinical trials of this combination of targeted agents in patients with relapsed neuroblastoma.

**Abstract:**

Activating mutations in the RAS/MAPK pathway are observed in relapsed neuroblastoma. Preclinical studies indicate that these tumors have an increased sensitivity to inhibitors of the RAS/MAPK pathway, such as MEK inhibitors. MEK inhibitors do not induce durable responses as single agents, indicating a need to identify synergistic combinations of targeted agents to provide therapeutic benefit. We previously showed preclinical therapeutic synergy between a MEK inhibitor, trametinib, and a monoclonal antibody specific for IGF1R, ganitumab in RAS-mutated rhabdomyosarcoma. Neuroblastoma cells, like rhabdomyosarcoma cells, are sensitive to the inhibition of the RAS/MAPK and IGF1R/AKT/mTOR pathways. We hypothesized that the combination of trametinib and ganitumab would be effective in RAS-mutated neuroblastoma. In this study, trametinib and ganitumab synergistically suppressed neuroblastoma cell proliferation and induced apoptosis in cell culture. We also observed a delay in tumor initiation and prolongation of survival in heterotopic and orthotopic xenograft models treated with trametinib and ganitumab. However, the growth of both primary and metastatic tumors was observed in animals receiving the combination of trametinib and ganitumab. Therefore, more preclinical work is necessary before testing this combination in patients with relapsed or refractory RAS-mutated neuroblastoma.

## 1. Introduction

Neuroblastoma is an embryonal tumor of the peripheral sympathetic nervous system. Neuroblastoma accounts for 6% of all cancers in children and 15% of the deaths in children due to cancer. Patients with high-risk neuroblastoma, including patients with metastatic disease at diagnosis, have an overall survival of less than 50%, despite aggressive multi-modality treatment, including chemotherapy, surgery, radiation, differentiation therapy, immunotherapy, and stem cell transplant. Up to 60% of patients with high-risk neuroblastoma relapse following up-front therapy, and long-term survivors have significant morbidities, indicating a need for additional therapeutic options, including molecularly targeted therapies [1]. 

The advent of next-generation sequencing (NGS) has facilitated the discovery of the genomic drivers of cancer development and, thus, relevant targets for drug discovery for many human malignancies. Despite significant effort, however, very few recurrent targetable mutations have been identified in high-risk neuroblastoma. Focal amplification of the oncogene *MYCN* and hemizygous deletion of chromosomes 1p and 11q are associated with high-risk disease [2]. Activating mutations in the receptor tyrosine kinase *ALK* and loss of function mutations or deletions of genes important for chromatin remodeling, including *ATRX, ARID1A*, and *ARID1B*, are observed in high-risk neuroblastoma [2]. While somatic variations in the genes of the RAS/MAP kinase pathway are rarely observed in high-risk neuroblastomas at diagnosis [2,3,4], variants that lead to increased RAS/MAP kinase activity are enriched in relapsed neuroblastomas [5]. These variants include gain-of-function mutations in the RAS isoforms *HRAS*, *KRAS*, and *NRAS*; gain-of-function mutations in the protein phosphatase *PTPN11*; and loss-of-function mutations and deletions in the RAS GTPase activating protein *NF1*. RAS/MAP kinase pathway alterations are associated with poor overall survival in neuroblastoma [4]. The presence of a RAS mutation in neuroblastoma cell lines introduces a functional dependency on RAS [6]. The RAS/MAP kinase pathway, then, is a potential therapeutic target for treating relapsed neuroblastoma.

The RAS/MAP kinase pathway consists of a cascade of serine/threonine kinases. In this cascade, GTP-bound and active RAS recruits RAF kinase to the plasma membrane, where it is activated. Active RAF phosphorylates and activates MEK1 and 2, which in turn phosphorylate and activate the MAP kinases, ERK1 and ERK2. ERK1 and ERK2 phosphorylate many additional substrates, which impact diverse cellular processes, such as proliferation, differentiation, migration, and survival. Initial efforts to target the RAS/MAP kinase pathway in neuroblastoma focused on the use of MEK inhibitors. As is the case in many other RAS-driven cancer models, MEK inhibitors as single agents have short-lived efficacy in preclinical models of neuroblastoma [5,7,8,9,10]. To improve the efficacy of MEK inhibitors in RAS-driven neuroblastoma, investigators have combined these agents with other molecularly targeted agents. The combination of a MEK inhibitor with a CDK4/6 inhibitor was synergistic in preclinical models of neuroblastoma [11], resulting in the initiation of a phase I clinical trial (NCT02780128). Combinations of MEK inhibitors with inhibitors of the PI3 kinase/AKT/mTOR pathway are also effective in preclinical models of neuroblastoma [12,13]. These combinations have been effective in models of other RAS-driven cancers, but their clinical success has been limited by toxicity [14]. Other promising combinations in RAS-driven neuroblastoma include a MEK inhibitor with retinoic acid [7,15,16], a CENPE inhibitor [17], a Hippo pathway modulator [18], a SHP2 inhibitor [19], and a CDK8 inhibitor [20]. Preclinical evaluations of combinations of a MEK inhibitor with a BET bromodomain inhibitor [21] or an ALK inhibitor [9] have been unsuccessful in neuroblastoma. 

In addition to the alterations in the RAS/MAPK pathway described above, most neuroblastoma tumors express IGF1R. The IGF1R pathway is aberrantly activated by autocrine and paracrine signaling loops in neuroblastoma tumors through the release of IGF1 and IGF2, ligands for IGF1R, from the neuroblastoma and stromal cells within the tumor [22,23,24,25]. The increased activity of the IGF1R/PI3 kinase/AKT/mTOR pathway induces proliferation, prevents apoptosis, increases motility, induces differentiation, and drives MYCN expression in neuroblastoma cells [26,27,28,29]. In addition, signaling through the IGF1R/PI3 kinase/AKT/mTOR pathway facilitates neuroblastoma metastasis to bone [30] and induces resistance to cytotoxic chemotherapy, retinoic acid, and ALK inhibitors [31,32,33]. Decreasing the expression of IGF1R [34] or AKT2 [12] in neuroblastoma cells decreases proliferation, migration, and invasion, validating the IGF1R/PI3 kinase/AKT pathway as a potential therapeutic target in neuroblastoma. Importantly, small-molecule and monoclonal antibody inhibitors of IGF1R potently decrease neuroblastoma cell viability and delay tumor growth in xenograft models of neuroblastoma [35,36,37,38,39,40,41,42,43,44]. Monoclonal antibodies specific for the IGF1R ligands [45], as well as inhibitors of PI3 kinase and mTOR, which are activated by signaling through IGF1R, are also effective in neuroblastoma models [46,47,48,49,50,51]. 

In our previous studies, we evaluated the efficacy of the combination of the MEK inhibitor, trametinib, with a monoclonal antibody specific for IGF1R, ganitumab, in RAS-mutated rhabdomyosarcoma [52]. Combinations of a MEK inhibitor and an IGF1R antibody have been effective in models of other RAS-mutated malignancies, such as lung cancer and leukemia [53,54]. In contrast to combinations of MEK inhibitors and PI3 kinase/mTOR/AKT inhibitors, the combination of a MEK inhibitor and an IGF1R antibody was well tolerated in adults [55]. Because both MEK and IGF1R inhibitors are effective in RAS-driven neuroblastoma, we sought to evaluate the combination of trametinib and ganitumab in this malignancy.

## 2. Materials and Methods

### 2.1. Cell Lines and Reagents

The neuroblastoma cell lines CHP-212 and LA-N-6 cell lines were obtained from the COG/ALSF Childhood Cancer Repository. SK-N-AS, NB-Eb-C1, NBL-S, SK-N-FI, and SK-N-BE(2)-C cells were obtained from the NCI Pediatric Oncology Branch. We ensured that our cells were free of mycoplasma contamination by testing with the MycoAlert kit (Lonza, Basel, Switzerland). In addition, we confirmed the identity of our cell lines by STR fingerprinting (Genetica/LapCorp, Burlington, NC, USA) (Appendix A). The neuroblastoma cell lines were grown in Roswell Park Memorial Institute 1640 medium (RPMI) with added fetal bovine serum (10% *v*/*v*), penicillin (100 IU/mL), streptomycin (100 mg/mL), and glutamine (2 mM). We obtained trametinib from the NIH Developmental Therapeutics Program (DTP) and ganitumab from the Cancer Therapy Evaluation Program (CTEP).

### 2.2. Whole-Exome Sequencing

Genomic DNA was extracted from neuroblastoma cell lines using Allprep DNA/RNA mini kits (Qiagen, Hilden, Germany). Sequencing libraries were prepared using the Agilent SureSelectXT human all exon V5 target enrichment kit (Agilent, Santa Clara, CA, USA). Paired-end 150 bp read sequencing was performed on a HiSeq3000 system using Illumina TruSeq V3 chemistry (Illumina, San Diego, CA, USA) at the CCR Sequencing Facility. Paired-end reads were mapped to the human genome (Hg38) using MWA-MEM 0.7.12 with default parameters. Data analysis was accomplished as previously described [52]. 

### 2.3. Cell Viability Assay

Cells were plated at a density of 10,000 cells/well in white, clear bottom 96-well plates. The next day, the cells were treated with trametinib at the indicated concentrations, with or without 1 µM ganitumab. The plates were incubated at 37  °C, with 5% CO2, for 72 h. To determine cell viability, 25 µL of CellTiter-Glo reagent (Promega, Madison, WI, USA) was added to each well. Luminescence was read on a SpectraMax iD3 plate reader (Molecular Devices, San Jose, CA, USA). The background signal from blank reactions (CellTiter-Glo reagent with cell culture media and no cells) was subtracted from the raw signal. The resulting values were normalized to DMSO control.

### 2.4. Matrix Combination Assay

The matrix combination assay was performed as previously described [52]. In brief, assay-ready plates were prepared by acoustic-droplet spotting of 25 nL DMSO-solvated trametinib and 1 µL RPMI-solvated ganitumab to each well. SK-N-AS cells were harvested and dispensed into the prepared plates to yield 500 cells/well. The plates were incubated at 37  °C, with 5% CO2, for 72 h, and then 2.5 µL CellTiter-Glo reagent (Promega) was dispensed into each well. We read the luminescence on a ViewLux instrument (Perkin-Elmer, Waltham, MA, USA). We normalized the resulting values to the DMSO control and performed the synergy calculations as previously reported [52].

### 2.5. Apoptosis Assays

For caspase 3/7 Glo assays, SK-N-AS or CHP-212 cells were plated at a density of 30,000 cells/well in white, clear-bottom 96-well plates. The next day, the cells were treated with vehicle (DMSO), trametinib, ganitumab, or trametinib/ganitumab. The plates were incubated at 37  °C, 5% CO2, for 18 h, and then 25 µL of Caspase 3/7-Glo reagent (Promega) was added to each well. Luminescence was read on a SpectraMax iD3 plate reader (Molecular Devices). The background signal from blank reactions (Caspase 3/7-Glo reagent with cell culture media and no cells) was subtracted from the raw signal. The resulting values were normalized to DMSO control.

For the Annexin V assays, SK-N-AS or CHP-212 cells were plated at a density of 500,000 cells/well in 6-well plates. The following day, the cells were treated with vehicle (DMSO), trametinib, ganitumab, or trametinib/ganitumab. The plates were incubated at 37  °C, 5% CO2, for 48 h. The resulting suspension cells were harvested, and the resulting adherent cells were detached with Accutase (Thermo, Waltham, MA, USA). The suspension and adherent cells were combined and incubated with APC-labeled human recombinant Annexin V and 7-Aminoactinomycin D (7-AAD) according to the manufacturers’ instructions (BD Biosciences, Franklin Lakes, NJ, USA). Samples were read on an Accuri C6 Plus flow cytometer (BD Biosciences). 

### 2.6. Capillary Immunoassays

Fresh frozen tumor samples were prepared in TPER (Thermo) using a TissueRuptor. Lysates were analyzed as described [56]. Primary antibodies included pERK (Cell Signaling Technologies # 9101, Danvers, MA, USA), total ERK (Cell Signaling Technologies # 9102), and IGF1R (Cell Signaling Technologies #3027)

### 2.7. Immunoblot Assays

Cells were treated as indicated, and lysates were analyzed essentially, as described in [56]. Primary antibodies include those detailed in Section 2.6, as well as NF1 (Bethyl # A300-140A), pS6 (Cell Signaling Technologies #2211), total S6 (Cell Signaling Technologies #2217), α-tubulin (Cell Signaling Technologies #2217), and vinculin (Sigma #h-vin1, Burlington, MA, USA). Secondary antibodies included anti-rabbit HRP (Cell Signaling Technologies #7074) and anti-mouse HRP (Cell Signaling Technologies #7076). 

### 2.8. Subcutaneous Xenograft Experiments

For both the subcutaneous and orthotopic xenograft studies, 6-week-old female SCID beige mice were purchased from Charles River laboratories. For the subcutaneous xenograft experiments, 2 million neuroblastoma cells (SK-N-AS or NB-Eb-C1) in a 1:1 mixture of Matrigel (BioTechne, Minneapolis, MN, USA) and HBSS (Sigma) were injected subcutaneously in the left flank of the mouse. Two weeks following tumor cell injection, the mice were randomized into treatment groups (*n* = 10 mice per group). No formal power calculation was performed prior to the start of these studies.

Trametinib suspensions were prepared in a vehicle, as previously described [52]. Trametinib was dosed in the indicated groups by oral gavage (OG) at the dosage reported previously [57]. Ganitumab was administered to indicated groups by intraperitoneal injection (IP), as reported previously [58].

In these experiments, the tumor dimensions were measured twice a week with digital calipers to obtain two diameters of the tumor sphere, from which the tumor volume was determined using the equation (D × d^2^)/6 × 3.14 (where D = the maximum diameter, and d = the minimum diameter). Two mice per group were euthanized 4 h after the fifth dose of vehicle or trametinib for assessment of pharmacodynamic markers of response, and the remaining mice were observed for tumor response. These animals were euthanized when they reached the tumor endpoint.

### 2.9. Orthotopic Xenograft Experiments

For the orthotopic xenograft experiments, intra-adrenal tumor cell injections were achieved via survival surgery. Mice (*n* = 20) were anesthetized with isoflurane and then placed laterally with the left flank facing upward. A lateral incision 0.5 inches below the spine on the animal’s right flank was made, followed by a small peritoneal incision to expose the kidney and visualize the adrenal gland. A Hamilton syringe was used to inject 3 × 10^5^ SK-N-AS cells in 7 uL of PBS. The kidney was then returned to the abdomen, the peritoneum was closed with absorbable suture, and the skin was closed with surgical staples. The staples were removed after 10–14 days. After the staples were removed, the mice were able to undergo ultrasound and MRI for tumor monitoring. Three weeks following tumor cell injection, the mice were randomized based on tumor volume and body weight into treatment groups (*n* = 10 mice per group). In these experiments, mice were euthanized when they gained more than 10% of their initial body weight due to tumor burden, or when the tumor volume exceeded 2000 mm^3^, as determined by ultrasound.

### 2.10. Mouse Imaging Preparation

Standard imaging and animal-handling protocols for both ultrasound and MRI required maintaining the rodent’s internal temperature and monitoring anesthesia administration. Animals were anesthetized at 3% isoflurane with a carrier gas of oxygen at a flow of 1 L/min in the induction chamber, and during imaging, the isoflurane was administered via a nose cone at 1.5–2.0%. Pulmonary function was monitored by a multichannel physiology system (MP150 Biopac System, Inc., Goleta, CA, USA), and the percent isoflurane administered via the nose cone was modified to maintain a pulmonary rate of 36–45 breaths per min (bpm) to reduce MRI motion artifacts. The animal body temperature was maintained by a thermostat-controlled heated table in the range of 34–37 °C during preparation, imaging, and post-imaging recovery.

### 2.11. Ultrasound

Ultrasound imaging was performed weekly after tumor cell injection, as previously described [59]. In brief, the area to be imaged was shaved with hair clippers, and additional hair was removed with depilatory cream (SurgiCream, American International Industries, Los Angeles, CA, USA), followed by 1% acetic acid. Heated gel (Aqua-Gel, Parker Laboratory, Inc., Fairfield, NJ, USA) was applied to match the tissue acoustic characteristics between the animal and the transducer, and the acoustic focus was placed at the center of the adrenal gland. B-mode images (Vevo2100, VisualSonics, Toronto, ON, Canada) were acquired using the 40 MHz transducer (MS-550S, VisualSonics, Toronto, ON, Canada), with an axial and lateral image spatial resolution of 40 and 90 µm, respectively. Three-dimensional volumes were calculated using vendor-supplied software (Vevo Lab v. 1.7.1, VisualSonics, Toronto, ON, Canada), utilizing the maximum axis linear X, Y, and Z measurements and the volume calculation (0.523 × (X × Y × Z)).

### 2.12. Magnetic Resonance Imaging

Magnetic Resonance Imaging (MRI) was performed weekly on a 3.0T clinical scanner (Philips Intera Achieva, Best, The Netherlands), as previously described [60], but with the following modifications. Custom-built volume receive array coils were utilized for simultaneously imaging of three mice to achieve high throughput. After a navigational survey scan with slices in sagittal, coronal, and axial view, a multi-slice T2 weighted turbo spin echo sequence (T2w-TSE) was applied. An 18 mm thick slab in coronal view was arranged to cover the whole mouse body. Image acquisition parameters: field of view, 160 × 78 mm^2^; flip angle, 90°; in-plane resolution, 0.18 × 0.18 mm^2^; slice thickness, 0.5 mm; repetition time (TR), 6000 ms; and echo time, (TE) 45 ms. A fat suppression imaging technique, Spectral Presaturation with Inversion Recovery (SPIR), was used to improve tissue contrast and facilitate the detection of metastases.

### 2.13. Electrochemiluminescence Assays

Neuroblastoma cells were lysed in Tris buffer with added protease and phosphatase inhibitors (Meso Scale Discovery, Rockville, MD, USA). The resulting lysates were used in the insulin signaling panel (total protein kit, Meso Scale Discovery, K15152C-1). The assay was read on a Meso Sector S600 (Meso Scale Discovery). Lysate from MCF-7 cells without growth factor stimulation (Meso Scale Discovery, Insulin Signaling Panel Whole Cell Lysate Set, C1151-1) was used as a positive control.

### 2.14. Histological Analysis

Tumor tissue was fixed in 10% neutral buffered formalin (NBF) and paraffin embedded. Paraffin-embedded sections of 5 µm thickness were prepared and stained with hematoxylin and eosin (H&E). The resulting slides were digitized with an Aperio ScanScope XT (Leica, Wetzlar, Germany) at 200× in a single z-plane. The whole-slide images were evaluated and annotated by a board-certified veterinary pathologist (EFE). 

## 3. Results

### 3.1. Drugs That Specifically Inhibit MEK1/2 and IGF1R Synergistically Inhibit Proliferation of RAS-Mutated Neuroblastoma Cells

The combination of the MEK inhibitor trametinib and the IGF1R monoclonal antibody ganitumab is effective in murine models of RAS-mutated rhabdomyosarcoma [52]. We hypothesized that this combination would be similarly effective in neuroblastoma with hyperactivity of the RAS/MAPK pathway, either through mutation of one of the RAS isoforms or through inactivation of the RAS GTPase activating protein, NF1. To test this hypothesis, we assembled a panel of RAS- and NF1-altered neuroblastoma cell lines (Figure 1A). A majority of these cell lines were derived from neuroblastoma tumors at relapse (SK-N-AS, LAN-6, NB-Eb-C1, SK-N-FI, and SK-N-BE(2)-C), with one cell line derived from a patient with newly diagnosed neuroblastoma (NBL-S) and one cell line for which the clinical information is unknown (CHP-212) [61]. We confirmed the RAS mutational status of the RAS-altered cell lines by whole-exome sequencing (Appendix A) and loss of NF1 expression at the protein level in NF1-altered cell lines via immunoblot (Figure 1B). We also confirmed that the cells expressed IGF1R at the protein level, using an electrochemiluminescence assay (Figure 1C) and immunoblot (Appendix A). These results established that IGF1R is highly expressed in SK-N-AS. In contrast, the expression of IGF1R in SK-N-FI cells is low. The remaining RAS/MAPK pathway-altered neuroblastoma lines, namely NB-Eb-C1, LAN-6, CHP-212, NBL-S, and SK-N-BE(2)-C, expressed IGF1R to a similar extent as the positive control cell line, MCF7. In contrast, all the RAS/MAPK pathway-altered neuroblastoma lines expressed the insulin receptor, which is highly homologous to IGF1R, to a similar extent as MCF7 cells, except CHP-212.

Next, we confirmed that the MEK inhibitor trametinib was potent and efficacious in this panel of RAS/MAPK-altered neuroblastoma cell lines. Trametinib was potent in all the RAS-altered neuroblastoma cell lines, with IC50 values for each cell line in the nanomolar range (Figure 2A, left). However, the efficacy of trametinib was poor in NB-Eb-C1, which showed a maximal effect of only 30% decreased viability. In contrast, trametinib was potent in only two of the three NF1-altered neuroblastoma cell lines (Figure 2B, right). SK-N-BE(2)-C was resistant to trametinib, while SK-N-FI and NBL-S had IC50 values in the nanomolar range. 

Despite these differences in the impact of trametinib on the viability in the RAS/MAPK pathway-altered neuroblastoma cell line panel, trametinib was able to decrease ERK phosphorylation in each of the cell lines studied (Figure 2B). Because they are viable despite decreased ERK phosphorylation in the presence of trametinib, SK-N-BE(2)-C cells may not be functionally dependent on the RAS/MAPK pathway for cell viability.

In contrast to previous reports using the IGF1R monoclonal antibody EM164 [37], the anti-proliferative effects of ganitumab as a single agent on neuroblastoma cells were modest (Appendix A). Ganitumab was most efficacious in CHP-212 cells, where it showed a maximal effect of 40% decreased viability. Ganitumab was most potent in SK-N-AS and NB-Eb-C1 cells, with IC50 values of 400 nM (Appendix A). These three RAS/MAPK-altered neuroblastoma cell lines were selected for further study based on their sensitivity to trametinib and ganitumab as single agents. To determine if the combination of MEK inhibition and IGF1R inhibition would synergistically decrease viability in RAS-mutated neuroblastoma cells, we performed a matrix combination assay in which the viability of SK-N-AS cells was determined in the presence of 10 different concentrations of either trametinib or ganitumab (Figure 3A). Synergy was observed between trametinib and ganitumab in SK-N-AS, according to the Bliss independence model. To assess if there was synergy in additional neuroblastoma cell lines, we tested the ability of a fixed concentration of ganitumab to affect the viability decrease achieved by trametinib alone. In NB-Eb-C1 cells, ganitumab did not alter the efficacy or potency of trametinib (Figure 3B, top); however, in CHP-212 cells, synergy with ganitumab was observed by a shift in the IC50 to the left in the presence of 1000 nM ganitumab (Figure 3B, bottom). We also assessed the impact of the combination of trametinib and ganitumab on apoptosis in RAS-altered neuroblastoma cells using both a luminescence-based assessment of caspase 3/7 activity (Figure 3C) and a flow cytometric assessment of annexin positivity (Figure 3D). Neither trametinib alone, ganitumab alone, nor the combination of trametinib and ganitumab increased caspase 3/7 activity or annexin positivity in SK-N-AS cells. However, trametinib in the presence and absence of ganitumab increased both caspase 3/7 activity and annexin positivity consistent with induction of apoptosis in CHP-212 cells (Figure 3C,D).

We used immunoblot experiments to confirm the on-target activity of trametinib and ganitumab at the doses used in our SK-N-AS apoptosis experiments (Figure 3E). After 3 h of treatment, 10 nM trametinib decreased ERK phosphorylation in SK-N-AS cells in the presence or absence of ganitumab; however, rebound ERK phosphorylation was observed in SK-N-AS cells treated with 10 nM trametinib for 48 h. Ganitumab at a dose of 1000 nM, both alone and in combination with trametinib, decreased total IGF1R expression in SK-N-AS after both 3 h and 48 h of treatment. Notably, the combination of trametinib and ganitumab decreased the phosphorylation of S6 ribosomal protein, a point of convergence of the MAPK and mTOR pathways [62], after 48 h of treatment. These results indicate that trametinib and ganitumab had on target effects in SK-N-AS.

### 3.2. Combined Trametinib and Ganitumab Treatment Is Efficacious in Murine Xenograft Models of RAS-Mutated Neuroblastoma

We next wanted to evaluate if ganitumab would provide therapeutic enhancement to trametinib monotherapy in xenograft models of RAS-mutated neuroblastoma. In SK-N-AS, consistent with prior reports [5], we observed a tumor growth delay for mice treated with either trametinib or ganitumab as single agents as compared to the mice that received the vehicle. We also observed a tumor growth delay for mice treated with the trametinib/ganitumab combination as compared to either single agent (Figure 4A, left). The tumor volumes on day 15 in mice treated with trametinib and ganitumab were smaller than those from mice treated with the vehicle or trametinib alone (Figure 4B, left). The tumor growth delay was associated with a survival advantage for the mice treated with the combination compared to trametinib alone but not ganitumab alone. Importantly, the trametinib/ganitumab combination was associated with the prolongation of overall survival compared to the vehicle (Figure 4C, left). Similarly, in NB-Eb-C1 xenografts, we observed a tumor growth delay (Figure 4A, right), decreased tumor volume on day 36 (Figure 4B, right), and prolongation of overall survival (Figure 4C, right) for the mice treated with the combination of trametinib and ganitumab compared to the vehicle. The phosphorylation of ERK (Figure 4D, left) and expression of IGF1R (Figure 4D, right) was decreased in tumor lysates from SK-N-AS and NB-Eb-C1 xenografts treated with trametinib and ganitumab, indicating that the in vitro mechanisms of action of each of these drugs was preserved in vivo. In summary, these results suggest that the addition of an IGF1R inhibitor to a MEK inhibitor provided modest therapeutic enhancement in heterotopic RAS-mutated neuroblastoma models. 

Adrenal SK-N-AS xenografts are more vascular, more locally invasive, and more metastatic as compared to subcutaneous SK-N-AS xenografts [63]. We hypothesized that the combination of trametinib and ganitumab would be more effective in adrenal than subcutaneous SK-N-AS xenografts because the orthotopic site (adrenal) is a better model of the human tumor microenvironment. To test this hypothesis, we injected unmodified SK-N-AS cells into the right adrenal gland of SCID beige mice and monitored tumor growth via once-weekly ultrasound and MRI imaging. When the adrenal tumors had engrafted as determined by ultrasound, we randomized the mice to receive either the vehicle or the trametinib/ganitumab combination. Similar to the subcutaneous xenografts, trametinib/ganitumab caused a tumor growth delay in adrenal SK-N-AS xenografts (Figure 5A). Upon necropsy for humane endpoints, the xenograft weight was decreased in the trametinib/ganitumab-treated group as compared to the vehicle-treated group (Figure 5B). However, the combination of trametinib and ganitumab was not associated with the prolongation of overall survival compared to the vehicle alone in adrenal xenografts (*p* = 0.0946, log-rank test) (Figure 5C). Surprisingly, we also observed that four of the nine mice (44%) in the trametinib/ganitumab group developed new nodules in the liver or lungs during treatment (Figure 5D). One of these nodules was confirmed by histopathology to be neuroblastoma (Figure 5E), consistent with metastasis. In contrast, only one of the nine mice (11%) in the vehicle group developed new nodules. The presence of metastatic disease could account for the lack of survival advantage conferred by trametinib/ganitumab for mice bearing adrenal tumors. 

## 4. Discussion

In this study, we showed that the MEK inhibitor trametinib and the IGF1R inhibitor ganitumab synergistically inhibited RAS-mutated neuroblastoma cell proliferation and induced apoptosis in cell culture models. The response to ganitumab was not associated with the expression level of IGF1R in these cells. The trametinib/ganitumab combination delayed tumor growth in both of the heterotopic cell line xenograft models tested and delayed tumor growth in an orthotopic cell line xenograft model of neuroblastoma. Interestingly, the trametinib/ganitumab combination did not prevent neuroblastoma metastasis in the orthotopic xenograft model. 

Several types of tumor cell resistance to molecularly targeted agents have been identified: intrinsic, adaptive, and acquired [64]. Intrinsic resistance refers to resistance that is present within tumor cells before drug exposure. Adaptive resistance refers to the changes occurring within tumor cells after short-term exposure to a targeted agent that can compensate for its action. In contrast, acquired resistance occurs in tumors that initially respond to the targeted agent and then regrow [64]. Intrinsic, adaptive, and acquired resistance to MEK inhibition in RAS- or NF1-altered cells are all caused by cellular changes that either result in hyperactivation of the MAPK pathway or activation of alternate signaling pathways, such as the IGF1R/PI3K/AKT pathway [65]. Both RAS mutations and NF1 deletion confer sensitivity to MEK inhibition [66]. However, here we confirmed that SK-N-BE(2)-C cells display intrinsic resistance to MEK inhibition despite demonstrating loss of NF1 expression, a result that has been shown previously [5]. Intrinsic resistance to MAPK pathway inhibitor in cell lines with loss of NF1 expression has also been seen in melanoma [67] and glioblastoma [68], suggesting that there are additional biomarkers of MEK inhibitor sensitivity that require elucidation. 

One approach toward the prevention of adaptive resistance is combining MEK inhibitors with molecularly targeted agents that either augment the inhibition of the RAS/MAPK pathway achieved by the MEK inhibitor (vertical pathway inhibition) or block signaling through the compensatory signaling pathways (horizontal pathway inhibition) [66]. In this study, we attempted to accomplish both vertical and horizontal pathway inhibition by combining a MEK inhibitor with an inhibitor of a receptor tyrosine kinase (RTK), IGF1R, which signals through both the RAS/MAPK pathway and a potential compensatory signaling pathway, the PI3K/AKT pathway. IGF1R is a functional dependency in neuroblastoma cells [69]. However, in our neuroblastoma xenografts, the trametinib/ganitumab combination delayed but did not prevent tumor growth, indicating the development of adaptive resistance. This resistance could be due to the activation of an additional RTK. Several RTKs, including ALK, KIT, MET, NTRK2, and RET, play a major role in neuroblastoma pathogenesis [70,71,72], and their activation could be responsible for adaptive resistance to trametinib and ganitumab in neuroblastoma. Notably, strategies to inhibit RAS activation downstream of RTKs, such as with an SHP2 inhibitor, as has been previously reported [19], would not completely prevent the activation of compensatory signaling pathways such as the PI3K/AKT [73] or JAK/STAT [74] pathways, which can be activated in a RAS-independent manner.

We also evaluated the impact of trametinib/ganitumab on tumor invasion in an orthotopic SK-N-AS xenograft. We did not assess for tumor invasion and metastasis in the heterotopic xenograft models. Previous studies have shown that orthotopic SK-N-AS xenograft tumors directly extend into the ipsilateral lobes of the liver [63]. We observed this direct extension, as well as the evidence of new liver and/or lung nodules, consistent with the development of distant metastases. These nodules developed during treatment with trametinib and ganitumab even though the primary tumors were smaller in mice receiving treatment. In fact, a higher percentage of mice receiving trametinib and ganitumab experienced nodule formation as compared to mice receiving vehicle. Our study was not designed to determine the individual effects of MEK inhibition and IGF1R inhibition on neuroblastoma tumor dissemination. We have not assessed the impact of trametinib and ganitumab on the dissemination of additional orthotopic xenograft models and cannot rule out the possibility that our observations are specific to SK-N-AS. Previous studies have shown that the genetic knockdown of IGF1R expression using shRNA decreased the transwell invasion of neuroblastoma cells such as IMR32 and SH-SY5Y [34]; however, the effect of IGF1R inhibition on RAS- or NF1-altered neuroblastoma cell invasion is unknown. Interestingly, in some melanoma cell lines, treatment with the MEK inhibitor selumetinib (AZD6244) increased invasion in a spheroid assay [75], but the effects of MEK inhibition on neuroblastoma cell invasion has yet to be determined. Future studies will define the effects of MEK or IGF1R inhibition on the invasive and metastatic capacity of RAS- and NF1-mutated neuroblastoma models.

## 5. Conclusions

In this study, we assembled a panel of RAS- and NF1-altered neuroblastoma cell lines with variable IGF1R expression to evaluate the efficacy of the combination of a MEK inhibitor (trametinib) and an IGF1R inhibitor (ganitumab) in neuroblastoma. We found that some neuroblastoma cell lines were resistant to single-agent MEK inhibition despite the presence of a RAS or NF1 alteration. In addition, we showed that trametinib and ganitumab inhibited proliferation in RAS-mutated neuroblastoma cells, even cells with a low expression of IGF1R. We further showed that the trametinib/ganitumab combination inhibited the growth of heterotopic and orthotopic neuroblastoma xenografts. However, the trametinib/ganitumab combination appeared to increase metastasis of a RAS-mutated neuroblastoma orthotopic xenograft, which is concerning and merits further investigation. This study highlights the importance of using orthotopic xenograft models to investigate the effects of targeted agents on metastasis preclinically.

## Figures and Tables

**Figure 1 cancers-16-02320-f001:**
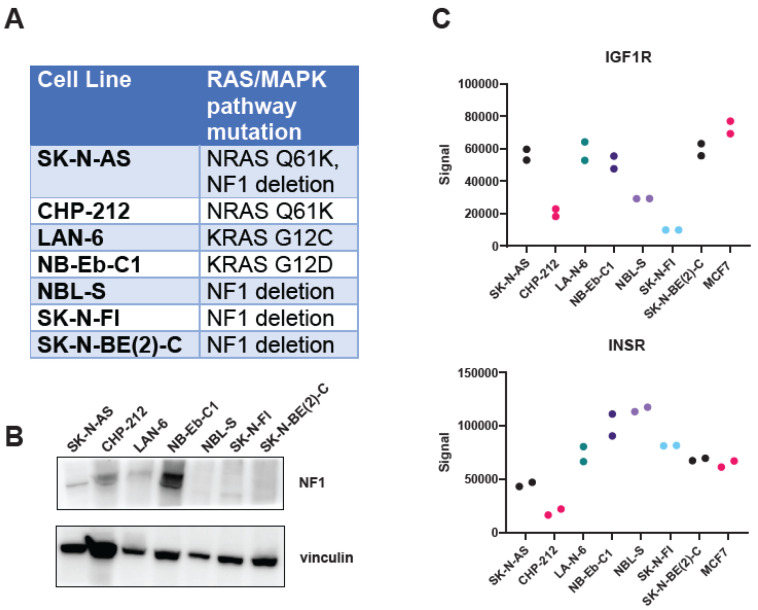
Neuroblastoma cell lines with alterations in the RAS/MAPK pathway express IGF1R. (**A**) RAS/MAPK pathway mutation status of the neuroblastoma cell line panel used in this study. (**B**) NF1 expression in the neuroblastoma cell line panel was determined by immunoblot. Vinculin immunoblot served as a loading control. (**C**) Expression of IGF1R (**top**) or the insulin receptor (InsR, **bottom**) in the panel of neuroblastoma cell lines was determined by an ECL-based sandwich immunoassay (MSD). MCF7 lysate was included as a positive control. Technical duplicates are displayed. Each cell line is represented by different color circles. The original immunoblot images can be found in the Appendix A.

**Figure 2 cancers-16-02320-f002:**
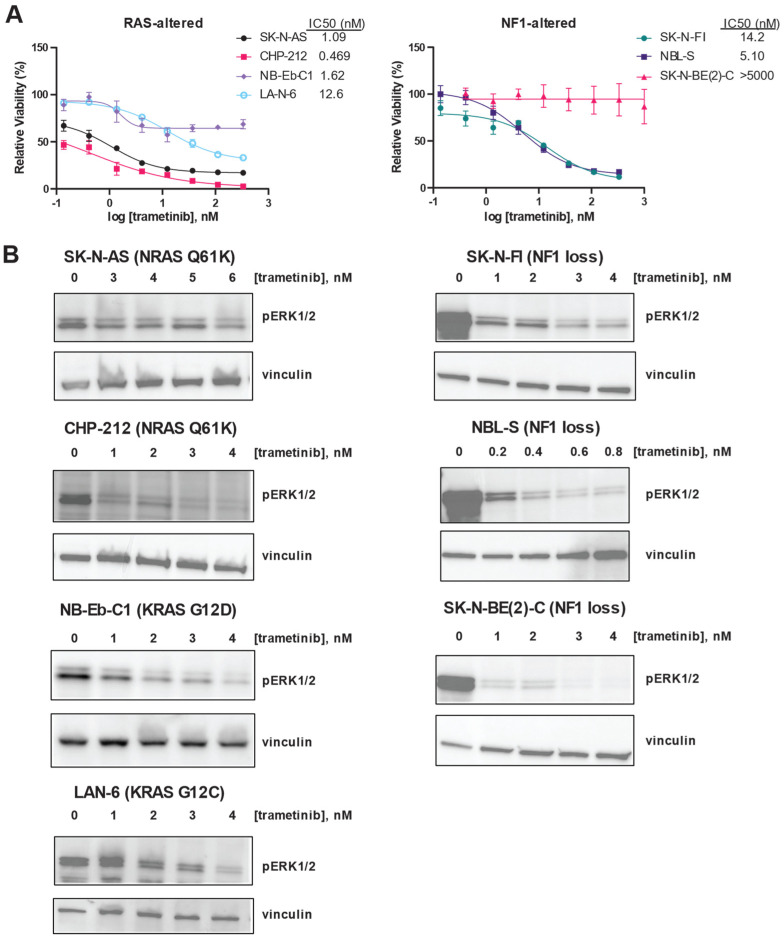
RAS mutation correlates with increased sensitivity to trametinib in neuroblastoma cells. (**A**) Efficacy and potency of trametinib were determined in RAS-altered (**left**) and NF1-altered (**right**) neuroblastoma cells using CellTiter-Glo signal 72 h after treatment as a marker of viability. Means of technical triplicates are displayed. Error bars indicate the standard deviation. (**B**) Erk1/2 phosphorylation after 24 h of treatment with the indicated concentrations of trametinib was determined by immunoblot in 4 RAS-altered neuroblastoma cell lines (**left**) and 3 NF1-altered neuroblastoma cell lines (**right**). The original immunoblot images can be found in the Appendix A.

**Figure 3 cancers-16-02320-f003:**
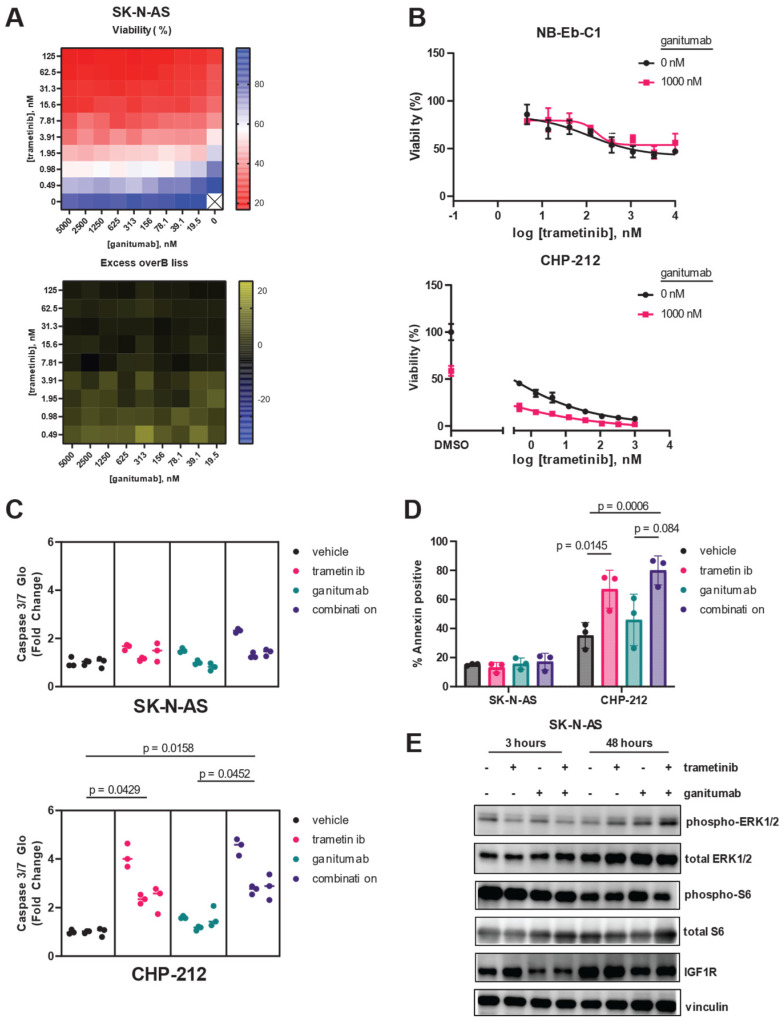
Trametinib and ganitumab synergistically inhibit RAS-mutant neuroblastoma viability. (**A**) Matrix (10 × 10) plot for the combination of trametinib (0 to 125 nM) and ganitumab (0 to 5000 nM) in both viability (CellTiter-Glo, **top**) and Excess-over-Bliss (**bottom**) format for SK-N-AS cells. (**B**) Efficacy of trametinib and the trametinib/ganitumab combination was determined in NB-Eb-C1 (**top**) and CHP-212 (**bottom**) neuroblastoma cells using CellTiter-Glo signal 72 h after treatment as a marker of viability. Cells were treated with varying concentrations of trametinib in the presence or absence of ganitumab (1 µM). Means of technical triplicates are displayed. (**C**) Caspase 3/7 activity of SK-N-AS (**top**) or CHP-212 (**bottom**) cells 18 h after treatment with vehicle, trametinib, ganitumab, or the combination of trametinib and ganitumab. SK-N-AS cells were treated with 10 nM trametinib; CHP-212 cells were treated with 4 nM trametinib. Both cell lines were treated with 1 µM ganitumab. Nested plots of three biological replicates (experimental units) comprising 3 technical replicates are displayed. Individual data points are shown; lines denote the median for that experimental unit. P-values were determined by nested 1-way ANOVA performed on transformed data. (**D**) SK-N-AS (**left**) or CHP-212 (**right**) cells were treated with vehicle, trametinib, ganitumab, or trametinib/ganitumab for 48 h. SK-N-AS cells were treated with 10 nM trametinib; CHP-212 cells were treated with 4 nM trametinib. Both cell lines were treated with 1 µM ganitumab. The treated cells were stained with Annexin V-APC and 7-AAD and analyzed by flow cytometry. The percentage of Annexin-positive cells is defined as the percentage of cells that were Annexin V positive and 7-AAD positive or negative. Individual biological replicates are shown. P-values were determined by 3-way ANOVA. (**E**) SK-N-AS cells were treated with vehicle, 10 nM trametinib, 1000 nM ganitumab, or the combination for 3 h (**left**) or 48 h (**right**), after which cells were harvested and the resulting cells were analyzed for phosphorylated and total forms of ERK, as well as IGF1R. Vinculin blot is included as a loading control. Representative blots are shown. The original immunoblot images can be found in the Appendix A.

**Figure 4 cancers-16-02320-f004:**
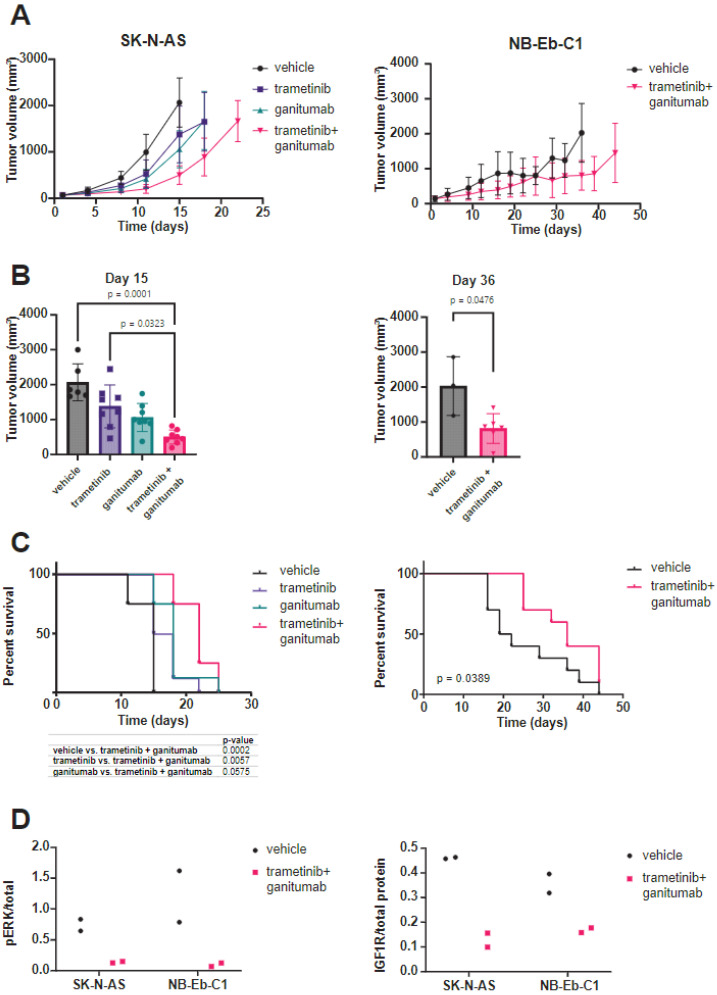
Trametinib and ganitumab delay tumor progression in heterotopic (subcutaneous) neuroblastoma cell line xenograft models. (**A**) SCID beige mice bearing 100–200 mm^3^ SK-N-AS (**left**) or NB-Eb-C1 (**right**) tumors were randomized to receive the indicated treatments (8 mice per group). Mean tumor volume ± one standard deviation is plotted. Means are displayed only if there are three or more tumor volume measurements at that time point. (**B**) Tumor volume at the study endpoint for mice bearing SK-N-AS (**left**) or NB-Eb-C1 (**right**) xenografts. P-values were determined by the Kruskal–Wallis test with Dunn’s correction for multiple comparisons for SK-N-AS and by the Mann–Whitney test for NB-Eb-C1. (**C**) Overall survival of mice bearing SK-N-AS (**left**) or NB-Eb-C1 (**right**) tumors treated as indicated. *p*-values determined by log-rank test. (**D**) Pharmacodynamic assessment was performed on tumors harvested from mice during treatment with vehicle or trametinib/ganitumab (4 h after the dose of vehicle or trametinib on the 5th day of treatment). The ratio of phosphorylated to total ERK and total IGF1R to total protein was determined by capillary immunoassay.

**Figure 5 cancers-16-02320-f005:**
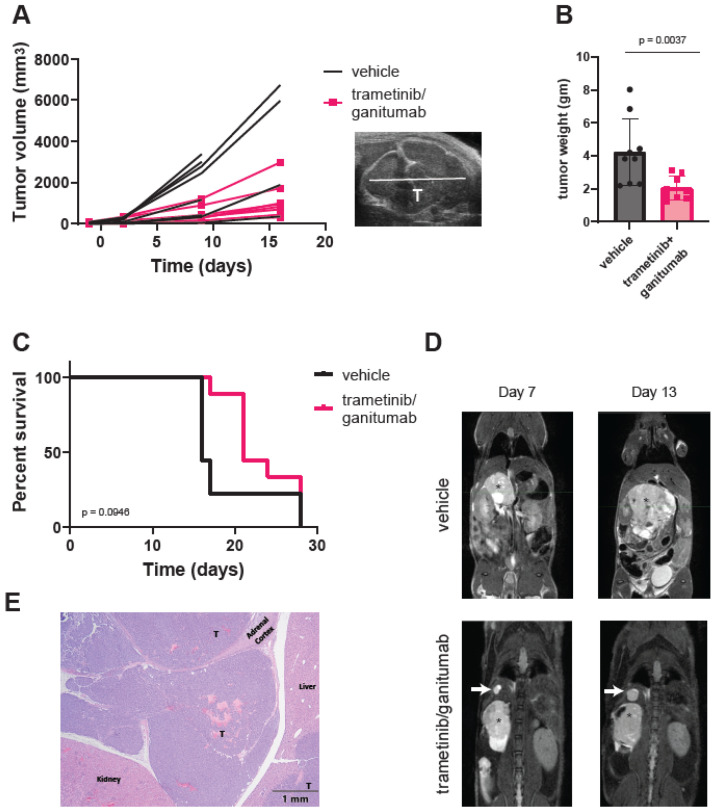
Trametinib and ganitumab delay tumor progression but do not prevent metastasis in an orthotopic (adrenal) SK-N-AS xenograft model. (**A**) Line graphs of tumor volume as a function of treatment duration for mice bearing adrenal SK-N-AS xenografts receiving vehicle or trametinib/ganitumab. Each line represents data from a single mouse (*n* = 9 per group). Tumor volume was estimated by ultrasound. A representative ultrasound image is shown (inset). “T” indicates the tumor. (**B**) Weight of xenografted tumors at study endpoint, as determined at necropsy. Bars indicate the mean; error bars indicate the standard deviation. *p*-value determined by Mann–Whitney test. (**C**) Overall survival of mice bearing SK-N-AS adrenal xenografts treated with either vehicle or trametinib/ganitumab. *p*-value determined by log-rank test. (**D**) Serial MRIs of SCID beige mice with SK-N-AS adrenal xenografts (indicated by an asterisk) receiving vehicle (**top**) or trametinib/ganitumab (**bottom**). White arrows indicate metastatic tumors. (**E**) Hematoxylin and eosin staining of the xenograft from the mouse treated with trametinib/ganitumab shown in panel (**D**). “T” denotes small, round blue tumor cells, consistent with neuroblastoma effacing the adrenal medulla and infiltrating into the adjacent peritoneal space. The upper-right liver lobe contains multifocal nodules of neoplastic cells within the hepatic parenchyma, one of which is also labeled with a “T”.

## Data Availability

The data generated in this study are available within the manuscript and its Appendix A. Sequencing data are available through dbGaP, accession number PRJNA1083467 (https://www.ncbi.nlm.nih.gov/sra/PRJNA1083467, accessed on 25 April 2024).

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
