# Peer review of "Preclinical Therapeutic Efficacy of RAF/MEK/ERK and IGF1R/AKT/mTOR Inhibition in Neuroblastoma"

_cancers, 2024, doi:10.3390/cancers16132320_

Round 1

Reviewer 1 Report

Comments and Suggestions for Authors

The combination of MEKi trametinib and IGF1R Ab ganitumab has exhibited a high efficacy against RAS-mutated rhabdomysarcoma in previous study.   In this study, Stauffer et al. determined whether this therapeutic could be extended to treat neuroblastoma with hyperactive RAS signaling since there’s no highly effective treatment for this aggressive cancer in clinic. They found that neuroblastoma cell lines with RAS/NF1 mutations were sensitive to the combination of Trametinib and Ganitumab in vitro, which was further validated in vivo by using xenograft murine models. Moreover, they demonstrated that this combination inhibited tumor growth although unable to block tumor metastasis in an orthotopic (adrenal) SK-N-AS xenograft model. Together, their data suggested that the combination of Trametinib and Ganitumab has some inhibitory effect on the development of neuroblastoma though more compelling data are still needed to support their conclusion. Overall, the quality of this manuscript is moderate, and needs to be improved significantly before considering for publishing.

1.        Fig1C, the expression of IGF1R and InsR needs to be verified by immunoblot.

2.        Fig2B, the quality of  immunoblots is low, particularly that for NB-EB-C1 cell line, needs to be improved.

3.        Fig3E, the combination of Trametinib and Ganitumab dose not exhibit any advantage against the ERK signaling in SK-N-AS cells, which is not consistent with authors’ conclusion.

4.        The observations in Fig5 is based on only single cell line, which needs to be verified by using at least one more cell line  in order to excluding cell line-specific effect.

Comments on the Quality of English Language

The English writing is good.

Author Response

  1.        Fig1C, the expression of IGF1R and InsR needs to be verified by immunoblot.

We have verified the expression of IGF1R by immunoblot. This experiment is shown in the new Supplemental Figure 1. The expression of InsR is not central to the manuscript and we have not completed that immunoblot.

2.        Fig2B, the quality of  immunoblots is low, particularly that for NB-EB-C1 cell line, needs to be improved.

The immunoblot for NB-Eb-C1 in figure 2B has been repeated, and the new experiment is now shown. 

3.        Fig3E, the combination of Trametinib and Ganitumab dose not exhibit any advantage against the ERK signaling in SK-N-AS cells, which is not consistent with authors’ conclusion.

Thank you for this comment. We initially hypothesized that we failed to see a decrease in ERK phosphorylation due to overexposure of the blot. However, we have repeated the experiment with a lower exposure and find that rebound ERK phosphorylation is still observed in the cells treated with a combination of ganitumab and trametinib. Based on the input from reviewer 2, we extended our analysis of the pathways altered by ganitumab and trametinib treatment to include phosphorylation of S6 ribosomal protein. We find that phosphorylation of S6 is decreased in the combination of trametinib and ganitumab. S6 ribosomal protein is a point of convergence between the RAS/RAF/MEK/ERK pathway and the IGF1R/PI3K/mTOR pathway, and decreasing S6 phosphorylation, rather than decreasing ERK phosphorylation, may be mediating the observed synergy between trametinib and ganitumab in neuroblastoma models. These results are shown in the new Figure 3E.

4.        The observations in Fig5 is based on only single cell line, which needs to be verified by using at least one more cell line in order to excluding cell line-specific effect.

We agree that we are unable to exclude a cell-line specific effect in Figure 5. We have added this point to lines 515-517 of the discussion.

Reviewer 2 Report

Comments and Suggestions for Authors

The manuscript investigates the efficacy of combining a MEK inhibitor (trametinib) and an IGF1R inhibitor (ganitumab) in neuroblastoma, focusing on RAS- and NF1-altered cell lines with variable IGF1R expression. The study demonstrates that while some neuroblastoma cell lines exhibit resistance to single-agent MEK inhibition, the combination of trametinib and ganitumab inhibits proliferation in RAS-mutated neuroblastoma cells, regardless of IGF1R expression levels. In preclinical xenograft models, although the combination therapy effectively delays tumor growth, it raises concerns about increased metastasis. Overall, the findings suggest that the combination of trametinib and ganitumab shows promise in inhibiting tumor growth. Additionally, the authors also mentioned that further research is needed to optimize therapeutic strategies for neuroblastoma treatment. However, several questions are still remained to be answered:

1. Additional experiments could be conducted to explore the potential mechanisms underlying the observed increase in metastasis in orthotopic neuroblastoma xenograft models. Specifically, investigating the molecular pathways or cellular processes underlying this phenomenon could provide valuable insights.

2. The efficacy of the inhbitors should be evaluated. Additionally, conducting knockdown experiments targeting specific genes or signaling pathways could help validate the findings.

3. A more comprehensive analysis of signaling pathways affected by single-agent or combined treatment with trametinib and ganitumab could be performed.

4. The results presented in Figure 4 should include statistical analysis to quantify the significance of observed differences between treatment groups.

5. The resolution of figures 2 through 4 appears to be suboptimal.

6. It is important to ensure consistency in reporting statistical values throughout the manuscript. The format of p-values should be standardized, either by presenting them as numeric values or using asterisks (*) to denote significance levels, to maintain clarity and consistency in data reporting.

7. In addition to histological analysis with H&E staining, immunohistochemistry could be performed to evaluate the expression levels and localization of specific molecular targets or signaling pathway components in tumor tissues.

Author Response

  1. Additional experiments could be conducted to explore the potential mechanisms underlying the observed increase in metastasis in orthotopic neuroblastoma xenograft models. Specifically, investigating the molecular pathways or cellular processes underlying this phenomenon could provide valuable insights.

We agree with this assessment, but feel these experiments are outside the scope of the current manuscript. These studies are suggested in lines 523-525 of the discussion. 

2. The efficacy of the inhibitors should be evaluated. Additionally, conducting knockdown experiments targeting specific genes or signaling pathways could help validate the findings.

The efficacy of trametinib as a single agent is shown in Figure 2A. We have added an experiment evaluating the efficacy of ganitumab as a single agent into a new Supplemental Figure 2. The efficacy of the combination of inhibitors is shown in Figure 3A and 3B. The knockdown experiments that the reviewer suggests are published. We now reference these studies in lines 107-109 in the introduction.

3. A more comprehensive analysis of signaling pathways affected by single-agent or combined treatment with trametinib and ganitumab could be performed.

Thank you for this suggestion. We have added phospho/total S6 ribosomal protein to this experiment now shown in Figure 3E.

4. The results presented in Figure 4 should include statistical analysis to quantify the significance of observed differences between treatment groups.

Statistical analysis of the changes in tumor volume at endpoint for the heterotopic models has now been included in Figure 4B.

5. The resolution of figures 2 through 4 appears to be suboptimal.

Apologies for this inconvenience. Figures with higher resolution have been included.

6. It is important to ensure consistency in reporting statistical values throughout the manuscript. The format of p-values should be standardized, either by presenting them as numeric values or using asterisks (*) to denote significance levels, to maintain clarity and consistency in data reporting.

The asterisks in Figure 3 have been replaced by numeric values.

7. In addition to histological analysis with H&E staining, immunohistochemistry could be performed to evaluate the expression levels and localization of specific molecular targets or signaling pathway components in tumor tissues.

We agree that IHC for pERK would be helpful. We have done the analysis of tumor tissue by capillary immunoassay, shown in Figure 4D.